# Efficiently Updating ECG-Based Biometric Authentication Based on Incremental Learning

**DOI:** 10.3390/s21051568

**Published:** 2021-02-24

**Authors:** Junmo Kim, Geunbo Yang, Juhyeong Kim, Seungmin Lee, Ko Keun Kim, Cheolsoo Park

**Affiliations:** 1Department of Computer Engineering, Kwangwoon University, Seoul 01897, Korea; wnsah1008@kw.ac.kr (J.K.); 2016722051@kw.ac.kr (G.Y.); kjoohyu@kw.ac.kr (J.K.); 2School of Electrical Engineering, College of Creative Engineering, Kookmin University, Seoul 02707, Korea; smlee27@kookmin.ac.kr; 3AI Lab, LG Electronics, Seoul 06763, Korea

**Keywords:** ECG, authentication, biometrics, incremental learning, SVM, incremental SVM

## Abstract

Recently, the interest in biometric authentication based on electrocardiograms (ECGs) has increased. Nevertheless, the ECG signal of a person may vary according to factors such as the emotional or physical state, thus hindering authentication. We propose an adaptive ECG-based authentication method that performs incremental learning to identify ECG signals from a subject under a variety of measurement conditions. An incremental support vector machine (SVM) is adopted for authentication implementing incremental learning. We collected ECG signals from 11 subjects during 10 min over six days and used the data from days 1 to 5 for incremental learning, and those from day 6 for testing. The authentication results show that the proposed system consistently reduces the false acceptance rate from 6.49% to 4.39% and increases the true acceptance rate from 61.32% to 87.61% per single ECG wave after incremental learning using data from the five days. In addition, the authentication results tested using data obtained a day after the latest training show the false acceptance rate being within reliable range (3.5–5.33%) and improvement of the true acceptance rate (70.05–87.61%) over five days.

## 1. Introduction

Biometrics has been widely applied in various consumer electronic products such as mobile phones and wearable devices to authenticate users with high performance [1,2]. For example, authentication systems based on fingerprint, iris, and face recognition are replacing passwords, given their reliability with adequate processing speed and better security than traditional authentication methods [3,4,5,6,7]. However, such biometric information is vulnerable to external attacks because common biometric features are physically exposed [8,9,10,11].

In contrast, electrocardiogram (ECG) signals, which exhibit unique physiological characteristics across subjects related to the placement and size of the heart, are difficult to spoof because the underlying biometric features are concealed during authentication and can only be obtained from physical measurements on the subject [12,13,14]. Since Biel et al. [15] first studied using ECG signal processing for the biometric recognition, ECG-based biometric authentication has received great attention as a next-generation promising technique and been implemented with various approaches to improve the authentication performance for the past few decades [16,17,18]. However, ECG signals of a person may vary according to his/her physical state or health condition, possibly leading to authentication failure in some cases [19,20,21].

Therefore, it is essential to design a robust method that handles the ECG intrasubject variability for accurate authentication. In order to achieve a reliable authentication result with the robustness to the non-stationarity of ECG, continuously recording and learning ECG data from the user can be one of the solutions. In this study, we applied incremental learning to efficiently and continuously update the user recognition model after every authentication attempt by the user. To the best of our knowledge, no study has considered incremental learning for ECG-based authentication.

Incremental learning performs continuous training as more input data become available to the existing model for extending its knowledge [22]. In recent years, large amounts of data or the data stream are produced constantly, thus raising the importance of efficiently training these data. The trained model can efficiently learn new patterns embedded in such data with the incremental learning while simultaneously preserving the previous model and preventing performing the retraining process from scratch. The main advantage of this method is that it can reduce the scale of the large size training set in the restricted memory and shorten the training time [23]. Furthermore, a system implementing incremental learning can accumulate knowledge throughout its life cycle. To date, incremental learning has been successfully integrated into various machine learning algorithms such as decision trees, neural networks, and support vector machines (SVMs) [24,25,26,27,28].

We propose a method for ECG-based biometric authentication with incremental learning of features. The method provides continuous training using new ECG signals as they become available and dynamically updates the existing authentication model while maintaining the previous authentication knowledge by implementing an incremental SVM [29].

The rest of this paper is organized as follows. In Section 2, we present the background and related works of ECG authentication and an incremental SVM algorithm for the proposed authentication method. The authentication scheme comprising acquisition and preprocessing of ECG signals, feature extraction, and incremental learning is detailed in Section 3. In Section 4, we report and analyze the experimental results. Section 5 provides a discussion of the proposed method and findings, and we draw conclusions in Section 6.

## 2. Background and Related Works

### 2.1. Electrocardiogram

Electrocardiogram, also known as ECG, is the record of electrical activity of the heart including depolarization and repolarization of the atrium and ventricle. ECG is composed of three fiducial entities: P wave, QRS complex, and T wave. The P wave and QRS complex are produced by the depolarization of the atrial and ventricle, respectively, and the T wave is produced by the repolarization of the ventricle. The basic waveform of ECG is shown in Figure 1. The ECG waveform varies from person to person owing to differences in the size and position of the heart, sex, age and other factors. Due to these characteristics, fiducial information such as angle, amplitude and interval of the entities of ECG describes the uniqueness of the individual [30].

### 2.2. Related Works

Table 1 presents several recent works on ECG authentication. Most of the ECG authentication methods used three well-known ECG databases: Physikalisch-Technische Bundesanstalt (PTB) [31], the Check Your Bio-signals Here initiative (CYBHi) [32] and MIT-BIH [33]. Some of the methods collected ECG data sets by themselves [15,34].

The extraction of features from ECG data can be classified into two categories: handcrafted and non-handcrafted. There are various handcrafted techniques for feature extraction such as using fiducial information [44,45,46], wavelet transform [40,47,48,49], and discrete cosine transform [42,50]. These approaches involve several processes such as feature normalization or removal of the noise designed by subjective decisions of the researchers. Thus, some researchers have implemented the authentication method with non-handcrafted extraction. With the advent of recent works of deep learning, non-handcrafted and data-driven feature extraction approaches using deep neural networks such as convolutional neural network (CNN) [36,51] or long short-term memory (LSTM) [52] can enable bypassing the additional steps.

Since a single biometric method cannot always guarantee high authentication performance [53], some studies adopted a multimodal-based method in order to provide more stable performance, such as a combination of ECG and fingerprint or photoplethysmography (PPG) [54,55,56].

### 2.3. Incremental SVM

The SVM is a supervised machine learning algorithm for classification and regression problems. The SVM classifier is trained using samples belonging to two or more categories and maximizes the margin around a decision boundary. As the SVM has demonstrated high performance without large training data sets, it has been widely used in a variety of applications, such as spam categorization [57], object recognition [58], and cancer localization [59]. However, the SVM training speed decreases rapidly as the size of the data set increases [60,61].

The incremental SVM extends the original SVM formulation to enable online learning while maintaining the previously trained model and efficiently updating it as more input data become available. As new data arrive, the conventional SVM should fully train all the existing data to reflect the latest information in the classifier. In contrast, the incremental SVM is retrained only using the new data, updating the model without complete retraining. Although the classification performance of the incremental SVM is similar to that of the conventional SVM, the incremental approach is more efficient in terms of training speed and number of floating-point operations during training [62].

The incremental SVM classifier is designed by solving a quadratic program on the Karush–Kuhn–Tucker (KKT) conditions to find the optimal solution:(1)$$\begin{array}{c} \alpha_{i}=0\to \left|f\left(x_{i}\right)\right|>\;1\\ 0<\alpha_{i}<C\to \left|f\left(x_{i}\right)\right|=\;1\\ \alpha_{i}=C\to \left|f\left(x_{i}\right)\right|<\;1 \end{array}$$
where $\alpha_{i}$ is the Lagrange multiplier for the constraint of the objective function, *f(x)* is the optimal separating function, and *C* is a regularization parameter that decides the degree of fitting to the training samples. If there is any violation of the KKT conditions during each incremental step, the coefficients of the samples change their values for the SVM classifier to keep satisfying the conditions. Hence, new samples are properly learned by the previously trained classifier.

For the proposed ECG-based authentication, we adopted the incremental SVM proposed by Cauwenberghs and Poggio [63]. This SVM manages data in four categories (Figure 2): set U of unlearned vectors that are going to be trained, set R of reserve vectors exceeding the margin, set S of margin support vectors within the margin, and set E of error support vectors violating the margin. During incremental learning, new training samples with *|f(x)|* > 1 (i.e., correctly classified) are assigned directly to set R, as they do not affect the SVM solution. Other samples initially become unlearned vectors in set U and are eventually assigned to set S or E. The KKT conditions must be simultaneously satisfied for all the training samples while integrating the unlearned samples into the SVM solution. The KKT conditions are maintained by modifying the coefficients of the margin vectors, and this process may result in the migration of samples to the other category in binary classification.

The incremental SVM stores training data because vectors other than the support vectors may become support vectors after the boundary update by incremental learning. As the number of trained samples increases, the memory required to store all the vectors also increases, thus prolonging training. To improve efficiency, the reserve vectors that are far from the decision boundary, and thus less likely to shift to the other category, may be discarded.

## 3. Proposed ECG-Based Authentication

The proposed authentication method and its evaluation are illustrated in Figure 3 and comprise acquisition and preprocessing of ECG signals, feature extraction, incremental learning, and performance evaluation.

### 3.1. ECG Signal Acquisition and Preprocessing

We recorded single-channel ECG signals from 11 male subjects aged from 22 to 42 for 10 min over six days. This experiment was conducted after obtaining the consent of all subjects and approved by Institutional Review Board (IRB). The recording and reference channels were placed on the left and right arms and the ground channel on the left ankle of the subject, forming the Einthoven’s triangle. The subject sat on a chair without moving during the ECG measurements. The MP36 system (Biopac Systems, Goleta, CA, USA) was used for the acquisition of the ECG signals at a sampling frequency of 1000 Hz. As the pre-processing step, we applied low-pass and high-pass filters to set a finite-impulse-response bandpass filter of order 5383 between 1 and 35 Hz [39,43,64] for the elimination of the baseline drift and high frequency noise.

The synchronized average of the ECG signals from each of the 11 subjects is shown in Figure 4, showing unique features in the ECG signals. Figure 5 illustrates the variation of the daily ECG waveform of the subject 10 for six days. Furthermore, the average root-mean-square error of ECG from days 2 to 6 with respect to the ECG of day 1 is presented in Table 2, showing the difference between the ECG of day 1 and those of days 2 to 6.

### 3.2. Feature Extraction

Typically, an ECG wave shows five fiducial points: P, Q, R, S and T peaks. Different combinations of peak patterns can be used as features to distinguish ECG data across subjects. Figure 6 and Figure 7 depict the peak detections of the QRS complex and P and T peaks. To automatically detect the Q, R, and S peaks, we used the real-time QRS detection algorithm proposed by Pan and Tompkins [65]. In addition, we separately extracted the P and T peaks by searching the maxima within
(2)$$R_{loc}-\frac{RR_{int}}{6}<P_{loc}<R_{loc}-\frac{RR_{int}}{10},$$
(3)$$R_{loc}+\frac{RR_{int}}{10}<T_{loc}<R_{loc}+\frac{RR_{int}}{2}$$
where $RR_{int}$ is the average length of RR interval, $R_{loc}$, $P_{loc}$ and $T_{loc}$ are the locations of R, P and T peak, respectively [66,67].

After the detection of five peaks, we extracted 13 features per ECG wave including three angles, four amplitudes, and six temporal features, which are illustrated in Figure 8: PR, QR, RS, RT amplitudes, PR, QR, RS, RT, ST, RR intervals, and angles of Q, R and S peaks [68].

### 3.3. Incremental Learning

In order to train the authentication model per subject, we obtained 11 initial incremental SVM classifiers that were designed using the corresponding training set of day 1. As the data set consisted of the authentication target (positive class) and 10 imposters (negative class), there was an imbalance between two classes, which degraded the authentication performance. In order to overcome this problem, the data augmentation for the positive class was conducted using the synthetic minority over-sampling technique (SMOTE) algorithm [69] in order to solve the imbalance. Each authentication model performs binary classification, either recognizing the target user’s ECG data or not.

After initial training, we used the data from days 2 to 5 for incremental learning. During this learning process, each training sample was evaluated by the existing SVM model to check its classification correctness. If the sample was positively classified, it became a reserve vector and the training terminated. Otherwise, margin vector coefficients were changed to maintain the KKT conditions in response to the perturbation induced by the misclassified sample. Then, the sample became either a margin vector or an error vector. After learning data incrementally, new information could be integrated into the existing SVM model without fully retraining on the complete training set.

### 3.4. Performance Measures

We used various evaluation measures to determine the authentication performance based on the true positives *TP*, false positives *FP*, true negatives *TN*, and false negatives *FN*. The true positives (true negatives) indicated the number of positive (negative) samples that were classified correctly. The false positives (false negatives) indicated the number of positive (negative) samples that were misclassified.

Specifically, we used three widely used evaluation measures, namely, accuracy *ACC*, false acceptance rate (*FAR*), and true acceptance rate (*TAR*):(4)$$ACC\;=\;\frac{TP+TN}{TP+TN+FP+FN}$$
(5)$$FAR\;=\;\frac{FP}{TN+FP}$$
(6)$$TAR\;=\;\frac{TP}{TP+FN}$$
where *FAR* reflected the incorrectly accepted attempts by an adversary, and *TAR* reflected the correctly accepted attempts by the authentication target.

## 4. Results

In this section, we designed two different evaluation schemes and tested our proposed ECG-based authentication algorithm, evaluated by using the data of day 6 and data recorded a day after the latest update as the test set. The evaluation based on the data of day 6 was conducted to compare the proposed algorithm with the conventional SVM. As this evaluation result could be affected by the data-to-data similarities, meaning the day 5 recordings would be more similar to the day 6 data than those of the day 1, we designed another experiment based on the latest available data to demonstrate the effectiveness of the incremental learning. In each evaluation, the test set consisted of data corresponding to one authentication target and 10 imposters (i.e., data from the other subjects). The evaluation proceeded until the classification of every single beat of the test set terminated. This process was repeated until the data from every subject were classified against those of all the other subjects. We evaluated the authentication method by an implementation on MathWorks MATLAB R2019b running on a Mac OS X 10.15 computer with Intel Core i7 CPU at 2.9 GHz and 16 GB memory.

In this paper, we used the radial basis function (RBF) as kernel function of the SVM, as it outperforms other kernel functions [70]. The hyperparameters of the SVM are box constraint *C* and kernel scale *σ*. The optimal hyperparameters should be determined before training the model to obtain the best performance. The search for the optimized values of the hyperparameters was conducted using Bayesian optimization, which was applied to the data recorded on day 1. This provided the optimal values of the hyperparameters as *C* = 7 and *σ* = 2, which are depicted in Figure 9 [71].

To visualize the updating decision boundary of the incremental SVM as more data became available, the two most representative features obtained from the ReliefF algorithm [72] (i.e., angle of S and RR interval) and the corresponding decision boundaries for subject 4 are depicted in Figure 10. The decision boundaries changed as incremental learning proceeded. As the incremental SVM integrated new data, the support vectors that defined the classification boundary were updated by varying their coefficients to maintain the KKT conditions.

Figure 11 shows the individual performance (a) and the average performance (b) evaluated using the data of day 6 in terms of the accuracy, FAR, and TAR of every subject over the five-day incremental learning. The initial classifier trained using the data of day 1 achieved 90.62% accuracy, 6.49% FAR and 61.32% TAR, on average. As the SVM classifier learned data incrementally on the other days, the authentication result showed enhanced performance: the accuracy increased to 95.1%, TAR increased to 87.61% and FAR decreased to 4.39%.

The experiment result over the five-day incremental learning as evaluated using the data recorded a day after the latest training is presented in Figure 12. Although there was a slight decrease of the performance when the evaluation was conducted using the data of day 5, the average accuracy across all subjects increased from 92.25% to 95.1%. The average TAR also gradually increased from 70.05% to 87.61%. The average FAR decreased from 5.33% to 3.5% until day 3, which was evaluated using the data of day 4. Then it slightly increased from 3.5% to 4.39% until the last incremental update.

## 5. Discussion

In Figure 12b, note that in terms of accuracy the authentication performance of FAR and TAR increased after the five days incremental learning. However, the accuracy slightly decreased when the evaluation was conducted using the data of day 5, and FAR increased about 0.89% from day 3 (tested using the data of day 4) to day 5 (tested using the data of day 6). Additionally, as seen in both Figure 11a and Figure 12a, some subjects had a minor decrease of authentication performance after training new ECG data incrementally. Since the pattern of the ECG signal could vary, depending on various reasons such as the emotional or physical conditions of the authentication target user, the pattern of the evaluated data might have been slightly different from the trained ones, resulting in the deterioration in the authentication performance. Nevertheless, FAR remained between 3% and 6% during the total training process, which indicated the model was still stable and reliable for the authentication task [73].

For a benchmark test, we compared the incremental SVM and the original SVM with full training in terms of the authentication performance and the training efficiency. Both SVMs with the RBF kernel were trained by the data of days 1 to 5 using the same hyperparameters described in Figure 9 (i.e., *C* = 7, *σ* = 2). The evaluation results tested using the data of day 6 are illustrated in Figure 13, showing that the accuracy of the incremental SVM was comparable with that of the fully trained SVM (95.1% and 95.12%). Likewise, FAR and TAR were similar between incremental learning and fully training methods: 4.39% and 4.4% FAR, 87.61% and 87.98% TAR, respectively.

Table 3 lists the training time of the incremental SVM and the original SVM. The training time was obtained while varying the number of arriving samples. Overall, incremental learning provided a much faster training than the fully training method. Specifically, incremental learning required 0.102–0.714 s to integrate one new sample, whereas the fully trained one required 0.713–3.294 s. When adding 100 new samples, the incremental training required 13.015–35.133 s and the other required 82.799–401.554 s for their computations. The training time across all subjects differed slightly in both methods since the SVM models had different numbers of the margin vectors and the error vectors and the training time of the SVM depended on the number of those vectors [29,61]. Table 3 shows that the difference in the training time between the two learning approaches increased as the number of new data increased. To integrate new information into the existing SVM model, the original SVM should repeat training using all the available data. On the other hand, the incremental SVM only modifies the coefficients of the margin vectors if an arriving sample is misclassified, thus reducing the training time efficiently.

In addition, the authentication method using the template update [74] was implemented and evaluated, which was similar to the proposed algorithm in terms of the continuous learning process. The evaluation was conducted by using the data obtained a day after the latest training as the test set. Figure 14 presents the comparison of FAR between the two methods, showing that our proposed algorithm obtained better performance than that using the template. Furthermore, as the model trained more data, the method using the template update had a problem of forgetting previous knowledge. However, incremental SVM maintained its knowledge by categorizing all training vectors in reserve, margin, or error sets.

Figure 15 shows the results of the proposed method and previous works [35,39] on ECG authentication using MIT-BIH and CYBHi databases. Using MIT-BIH, the proposed method obtained 97.7% accuracy, which was comparable (only 1% lower) with [35], although our method used seven fewer features. Furthermore, the proposed method yielded an accuracy of 99.4% using CYBHi, an almost identical performance with [39] (0.1% higher).

Table 4 details the training information of the authentication models, including the average number of training data samples, the proportions between the two classes (authentication target vs. others), and the training time of the incremental SVM per day. Although the number of arriving samples was similar over the five days, the training time significantly increased as more data became available and was stored, given the time required to calculate the kernel matrix of all the data [75]. Furthermore, the more data the incremental SVM learned, the more memory the authentication model occupied. Thus, our further work will focus on reducing the training time and memory by discarding some reserve vectors that are far from the decision boundary of SVM classifier.

## 6. Conclusions

We proposed an ECG-based authentication method providing incremental learning from arriving ECG data as they became available. The proposed algorithm was compared with the conventional ones, the original SVM and template update method. Compared to the original SVM, the proposed algorithm yielded almost identical performance with much a faster training process. In addition, it was demonstrated that the conventional continuous learning algorithm, the template update method, was outperformed by the proposed incremental learning approach. The proposed method was also compared with the previous studies on ECG authentication using MIT-BIH and CYBHi databases. The proposed algorithm achieved an accuracy of 97.7% and 99.4% using MIT-BIH and CYBHi, respectively, showing that the method could be as reliable as the others. We showed that our proposed algorithm could be reliable authentication method in terms of FAR and TAR, with the advantage of training new data incrementally. To the best of our knowledge, this is the first ECG-based authentication method implementing incremental learning, which is suitable to applications with data accumulation.

## Figures and Tables

**Figure 1 sensors-21-01568-f001:**
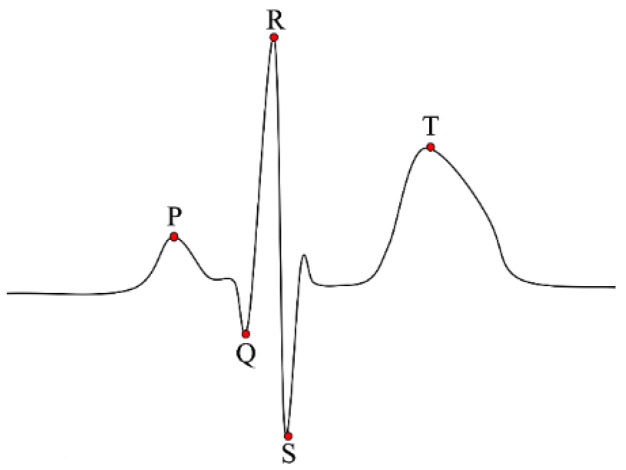
The basic waveform of electrocardiograms (ECG) with five fiducial points.

**Figure 2 sensors-21-01568-f002:**
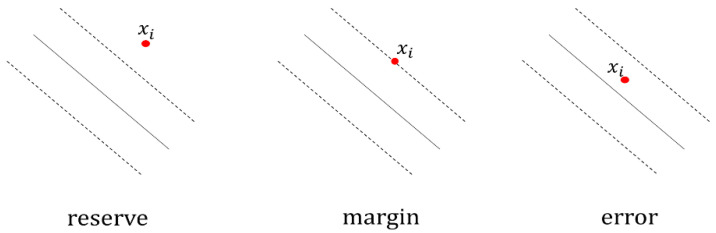
Sample states according to Karush–Kuhn–Tucker (KKT) conditions.

**Figure 3 sensors-21-01568-f003:**
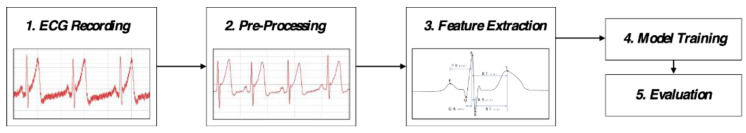
Scheme to implement and evaluate the proposed ECG-based authentication.

**Figure 4 sensors-21-01568-f004:**
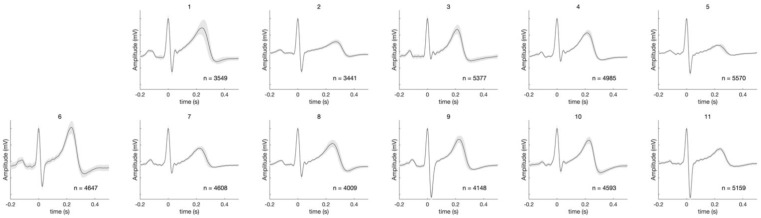
Synchronized average of ECG signals from 11 subjects. The shaded areas represent the standard deviation in the signals.

**Figure 5 sensors-21-01568-f005:**
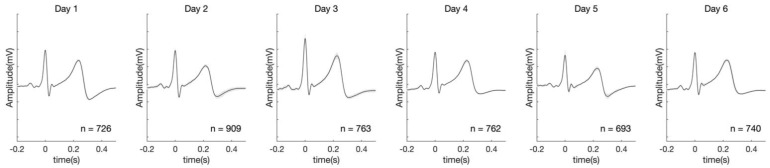
Synchronized average of ECG signals of the subject 10 for six days. The shaded areas represent the standard deviation in the signals.

**Figure 6 sensors-21-01568-f006:**
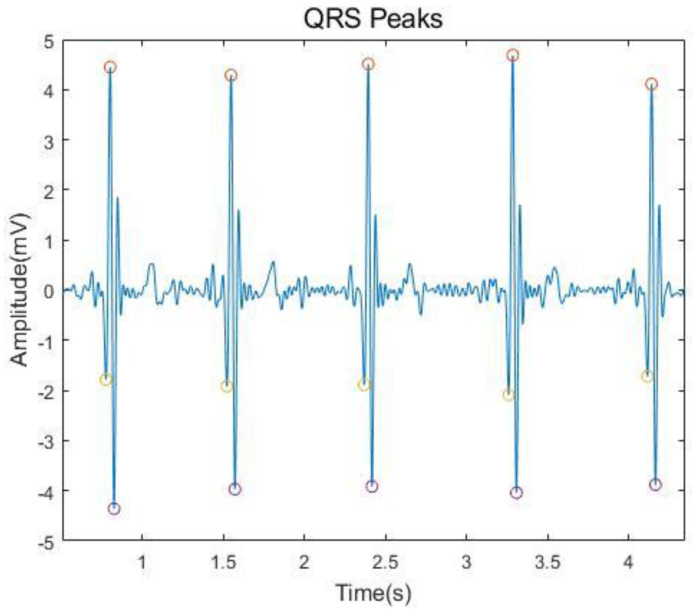
Detected Q, R and S peaks of an ECG signal.

**Figure 7 sensors-21-01568-f007:**
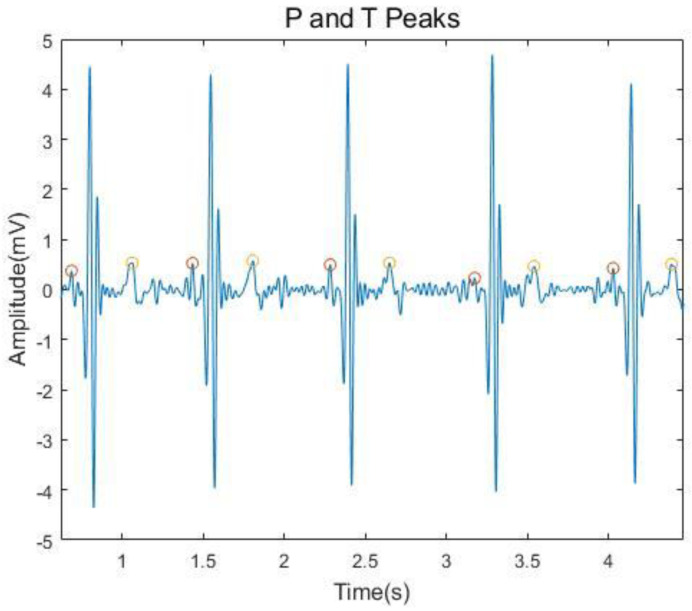
Detected P and T peaks of an ECG signal.

**Figure 8 sensors-21-01568-f008:**
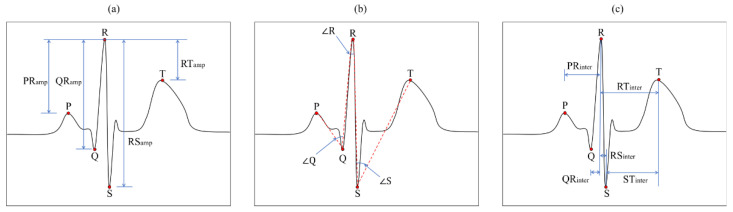
ECG features for the authentication: four amplitudes (**a**), three angles (**b**), and six temporal features (**c**) including RR interval.

**Figure 9 sensors-21-01568-f009:**
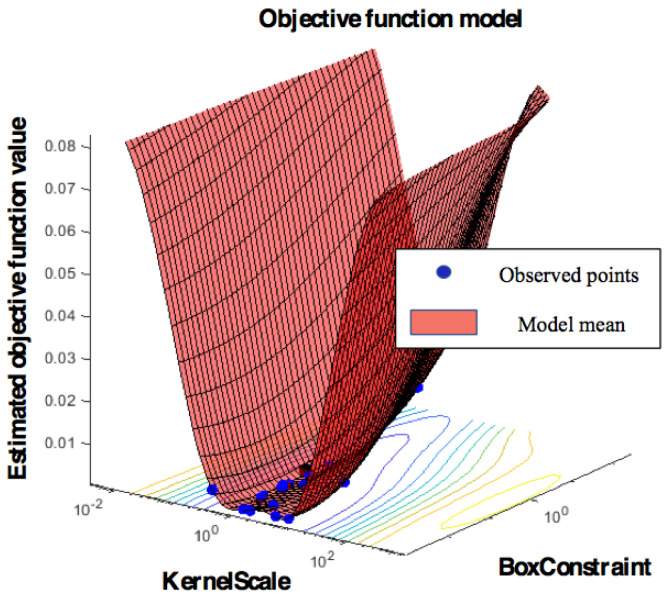
Objective function values for hyperparameter tuning using Bayesian optimizer. The optimal SVM parameters are *C* = 7 (box constraint) and *σ* = 2 (kernel scale).

**Figure 10 sensors-21-01568-f010:**
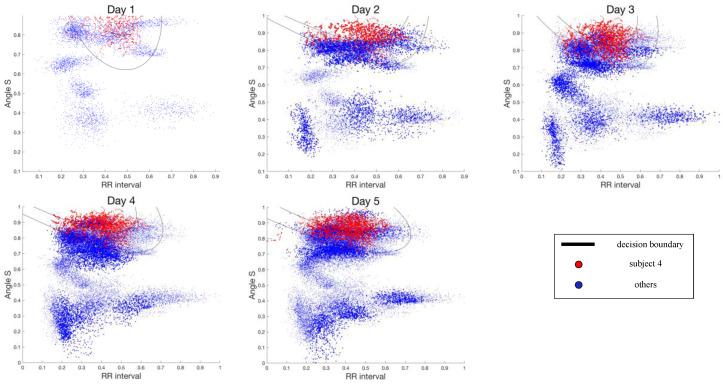
Updating decision boundary of incremental SVM for subject 4 data set over five-day incremental learning. The darker dots from days 2 to 5 represent the incrementally trained data.

**Figure 11 sensors-21-01568-f011:**
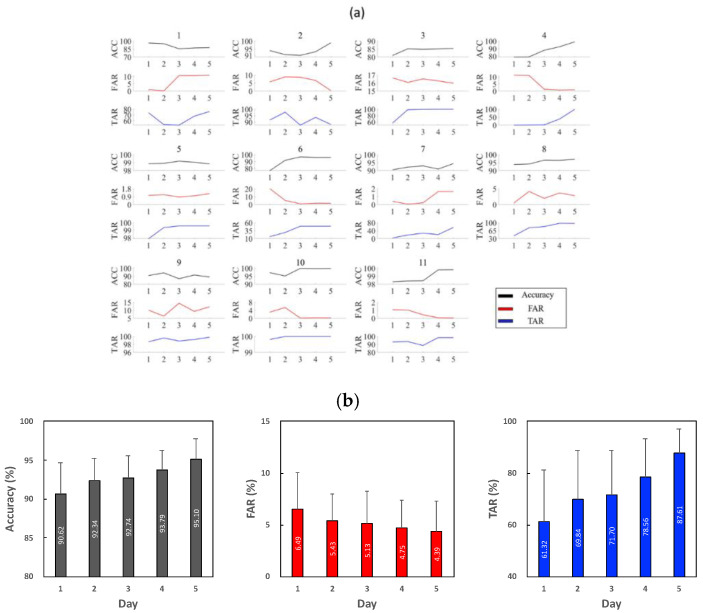
The accuracy, false acceptance rate (FAR) and true acceptance rate (TAR) of 11 subjects over five days of incremental learning (**a**) and the average (**b**) tested using the data of day 6.

**Figure 12 sensors-21-01568-f012:**
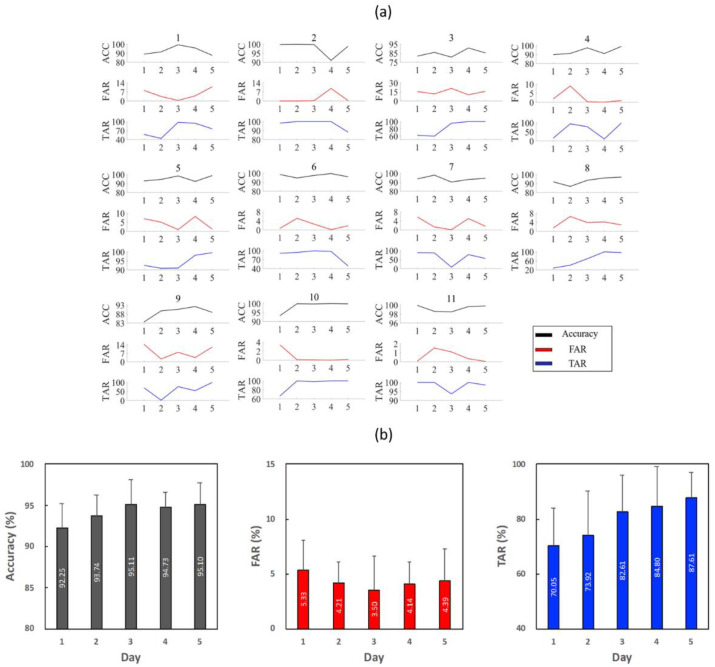
The accuracy, FAR and TAR of 11 subjects over five days of incremental learning (**a**) and the average (**b**) tested using the data recorded a day after the latest training.

**Figure 13 sensors-21-01568-f013:**
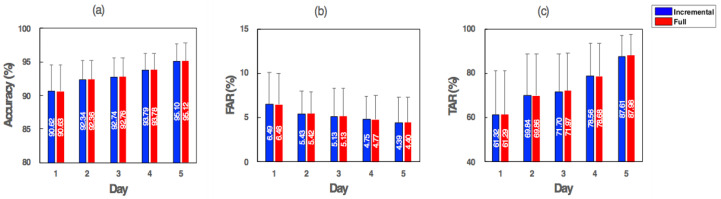
The result of the benchmark test with the incremental learning and full training in terms of (**a**) the accuracy, (**b**) FAR and (**c**) TAR.

**Figure 14 sensors-21-01568-f014:**
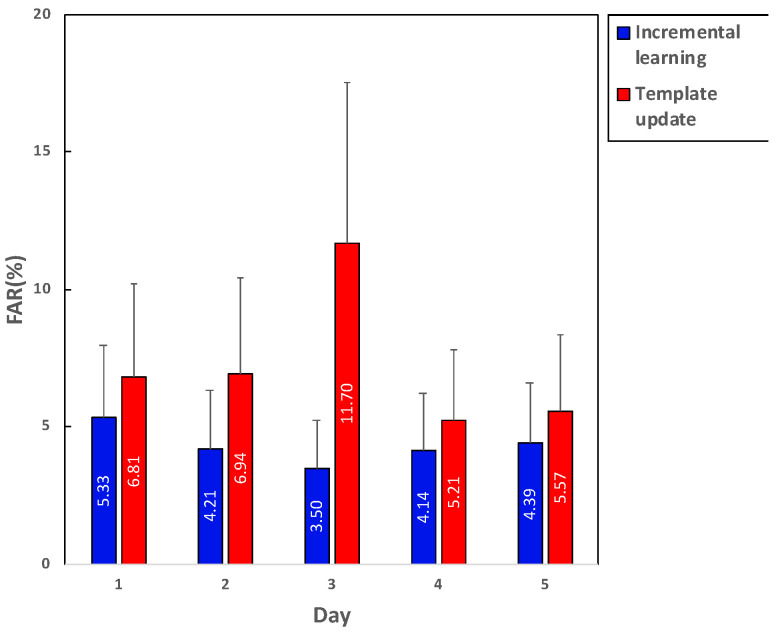
The result of the benchmark test with the incremental learning and template update in terms of FAR.

**Figure 15 sensors-21-01568-f015:**
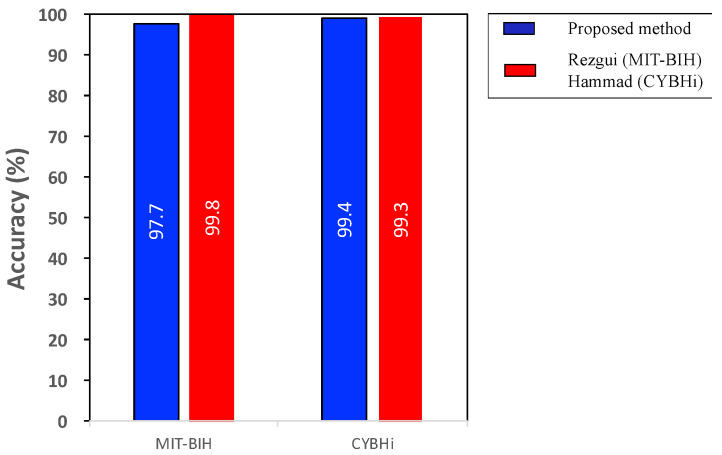
The evaluation result of the proposed method using MIT-BIH and Check Your Bio-signals Here initiative (CYBHi) databases. The experiment using MIT-BIH is compared with [35] and the other using CYBHi with [39].

**Table 1 sensors-21-01568-t001:** Summary of the studies of the ECG-based authentication method. SVM—support vector machine, ACC—accuracy, CNN—convolutional neural network, DWT—discrete wavelet transform, FAR—false acceptance rate, PSD—power spectral density, DCT—discrete cosine transform, PCA—principal component analysis.

Authors	Database	Feature Extraction	Authentication Model	Performance	
Rezgui et al. [35]	MIT-BIH	21 Fiducial and 10 morphological descriptors	SVM	ACC:	98.8%
Labati et al. [36]	PTB	CNN	Softmax	ACC:	100%
Juan et al. [20]	PTB self-collected	8 Fiducial	Template Matching	FAR:	1.29% 1.41%
Choi et al. [37]	MIT-BIH PTB	8 Fiducial	SVM	ACC:	95.9%
Ergin et al. [38]	MIT-BIH PTB	Fiducial, WT and PSD	Decision tree and BayesNet	F1-score:	0.972
Hammad et al. [39]	PTB CYBHi	ResNet-Attention	Softmax	ACC:	98.85% 99.27%
Chiu et al. [40]	QT DB [41]	DWT	Euclidean distance	ACC:	81%
Biel et al. [15]	Self-collected	Siemens ECG apparatus	PCA	ACC:	98%
Pinto et al. [42]	Self-collected	DCT	SVM	ACC:	94.9%
Hammad et al. [43]	MIT-BIH	Fiducial	CNN	ACC:	99%

**Table 2 sensors-21-01568-t002:** The average root-mean-square error of the ECG from day 2 to day 6 with respect to the ECG of day 1.

**RMSE**	**Day 2**	**Day 3**	**Day 4**	**Day 5**	**Day 6**
	0.12346 ± 0.06493	0.15085 ± 0.09276	0.16848 ± 0.07744	0.25785 ± 0.30352	0.20714 ± 0.14167

**Table 3 sensors-21-01568-t003:** Training time (s) using the incremental learning and full training while varying the number of arriving samples.

Subject	Incremental Learning			Full Training		
	1 Sample	10 Samples	100 Samples	1 Sample	10 Samples	100 Samples
1	0.541	3.995	27.184	2.022	20.279	241.554
2	0.630	3.233	25.382	0.713	7.101	82.799
3	0.213	3.320	17.190	2.570	25.502	322.653
4	0.367	5.661	35.133	1.225	12.762	192.827
5	0.289	4.347	34.006	1.901	18.343	229.996
6	0.234	2.116	13.015	1.306	13.181	161.457
7	0.252	3.208	26.130	2.185	21.436	262.371
8	0.102	2.332	20.584	1.497	15.139	187.970
9	0.156	1.994	17.990	3.294	31.457	401.554
10	0.610	4.711	28.913	0.951	9.294	107.644
11	0.714	3.381	21.442	1.120	9.227	113.987

**Table 4 sensors-21-01568-t004:** Averaged characteristics of training data and training time of the incremental learning. R, S, and E indicate the number of reserve vectors, margin support vectors, and error support vectors, respectively.

**Day**	Positive (%)	Negative (%)	Total	Training Time (s)	*R*	*S*	*E*
1	51.79 ± 6.2	48.21 ± 6.2	10,930 ± 1142	85.867 ± 25.6	10,348 ± 1059	37 ± 16	544 ± 420
2	51.14 ± 10.3	48.86 ± 10.3	10,525 ± 1408	157.070 ± 92.2	19,966 ± 1891	50 ± 23	1438 ± 1155
3	52.03 ± 4.65	47.97 ± 4.65	10,893 ± 914	344.520 ± 263.3	29,739 ± 2325	62 ± 33	2546 ± 1966
4	52.21 ± 3.19	47.79 ± 3.19	11,752 ± 653	525.066 ± 412.5	40,396 ± 2934	81 ± 28	3623 ± 2757
5	50.30 ± 14.49	49.70 ± 14.49	10,254 ± 1660	569.601 ± 582.5	49,808 ± 4057	85 ± 33	4461 ± 3367

## Data Availability

The data are not publicly available due to ethical.

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
