# Peer review of "Efficiently Updating ECG-Based Biometric Authentication Based on Incremental Learning"

_sensors, 2021, doi:10.3390/s21051568_

Round 1
Reviewer 1 Report
First of all, indeed, this research domain has recently became very intensively explored hence it is not easy to devise a method of ECG-based authentication that would be original in the first place. The Authors managed to come up with quite original method. This is worth emphasising. However, once the method originality is put aside, what is left is just another paper dealing with the ECG-based authentication. 70 papers long references only confirm how many publications related to this topic has recently appeared. I have no doubt (based on the included results) that the Authors have done a very detailed investigation of how their method works and what are its performance metrics. The problem is that they failed to present the results in the clear and easily comprehensible way. There are quite a few presentation-related issues, like figures 3, 4, 5, 8, 10, 11, 12, 13 as well as tables e.g. 3 and 4 are strangely aligned compared to the rest of the text. Then, figures 11a and 12b are too tiny to consider them useful. Maybe less but bigger or splitting this into some more and better readable figures would do some more justice to the results presented.
In terms of the results - objectively, there is evidence of performing a very thorough investigation which resulted in providing a method guaranteeing very good accuracy. On the other hand, as far as the accuracy is concerned, the proposed method does not significantly outperform the alternative methods. But it is worth noting that the training times have been noticeably reduced. The processing power required for the proposed methodology is quite substantial which raises questions regarding the potential application domain (at the first glance it seems not suitable e.g. for wearable devices).
And last but not least, the conclusion is very brief as for a paper that is 17 pages long. Even if that was due to moving discussion of the results to another part of the paper.
Use of English does not leave anything to be desired.
Overall, in the current form (especially due to presentation-related issues) the paper is not suitable for publication and would require quite a substantial revision.
Reviewer 2 Report
The papers contains many grammatical errors and needs to be proofread by and English expert. Some of these errors are detected by MS-Word. The paper lacks a more detailed introduction of the previous algorithms with emphasis on the differences, disadvantages and advantages of the proposed algorithm and each one of them, especially, the algorithms in [30,32,44]. The comparison between batch SVM and incremental SVM using the dataset in the paper can be removed since their findings are well established (Table 3 and the corresponding text). Instead, and much more important, the results should include a comparison to methods [30,32,44] either using the dataset that the the authors acquired or using the datasets in [30,32,44] (since [30,32,44] use not-incremental SVM). There is a mismatch between ref [30] in Table 1 and its content in the bibliography. There is a redundant underscore in "the maxima" in page 7.
Reviewer 3 Report
The paper presents a new method for biometric classification based on the ECG signal. The authors reported good results; however, some issues have to be better explained.
My main concern relates to the usability of the ECG recording for such an application. Is it convenient enough for prospective users?
The presented method should also be described more precisely.
1. What is the difference between the six day of the experiment and a day after the latest update as the test set?
2. Based on which data, the average root‐mean‐square error was calculated? Which values were taken into account for its calculation?
3. From which operations did the Pre-Processing step consist?
4. the training and testing vectors should be explained in more details, maybe graphically.
5. It is difficult to assess the values mentioned in the text, in Figures 4 and 5, parts (b). These values should be added at the top of each bar.
6. Why is there data for participant 4 presented in table 4 and not averaged for all participants?
7. The results should also be discussed regarding those presented in Table 1
8. The size of the figures should be corrected.
Round 2
Reviewer 2 Report
Th authors acknowledged almost fully the suggestions from the first review.
The paper can be accepted in its current form.
Reviewer 3 Report
Thank you for your answers.